# Assessment of Genotoxicity in Human Cells Exposed to Modulated Electromagnetic Fields of Wireless Communication Devices

**DOI:** 10.3390/genes11040347

**Published:** 2020-03-25

**Authors:** David Schuermann, Christina Ziemann, Zeinab Barekati, Myles Capstick, Antje Oertel, Frauke Focke, Manuel Murbach, Niels Kuster, Clemens Dasenbrock, Primo Schär

**Affiliations:** 1Department of Biomedicine, University of Basel, Mattenstrasse 28, CH-4058 Basel, Switzerland; zbarekati@gmail.com (Z.B.); f.focke@dkfz.de (F.F.); primo.schaer@unibas.ch (P.S.); 2Fraunhofer Institute for Toxicology and Experimental Medicine ITEM, Nikolai-Fuchs-Strasse 1, D-30625 Hannover, Germany; antje.oertel@item.fraunhofer.de (A.O.); clemens.dasenbrock@item.fraunhofer.de (C.D.); 3IT’IS Foundation, Zeughausstrasse 43, CH-8004 Zurich, Switzerland; capstick@itis.swiss (M.C.); manuel@murbach.eu (M.M.); kuster@itis.swiss (N.K.); 4Department of Information Technology and Electrical Engineering, Swiss Federal Institute of Technology (ETH), CH-8092 Zurich, Switzerland

**Keywords:** electromagnetic fields, DNA repair, DNA damage, genomic instability, genotoxicity, environment and public health, UMTS, GSM, WiFi, RFID

## Abstract

Modulated electromagnetic fields (wEMFs), as generated by modern communication technologies, have raised concerns about adverse health effects. The International Agency for Research on Cancer (IARC) classifies them as “possibly carcinogenic to humans” (Group 2B), yet, the underlying molecular mechanisms initiating and promoting tumorigenesis remain elusive. Here, we comprehensively assess the impact of technologically relevant wEMF modulations on the genome integrity of cultured human cells, investigating cell type-specificities as well as time- and dose-dependencies. Classical and advanced methodologies of genetic toxicology and DNA repair were applied, and key experiments were performed in two separate laboratories. Overall, we found no conclusive evidence for an induction of DNA damage nor for alterations of the DNA repair capacity in cells exposed to several wEMF modulations (i.e., GSM, UMTS, WiFi, and RFID). Previously reported observations of increased DNA damage after exposure of cells to GSM-modulated signals could not be reproduced. Experimental variables, presumably underlying the discrepant observations, were investigated and are discussed. On the basis of our data, we conclude that the possible carcinogenicity of wEMF modulations cannot be explained by an effect on genome integrity through direct DNA damage. However, we cannot exclude non-genotoxic, indirect, or secondary effects of wEMF exposure that may promote tumorigenesis in other ways.

## 1. Introduction

Data transmission by modulated radiofrequency electromagnetic fields (RF-EMFs) has become an integral part of our daily life. The ever-increasing use of wireless communication technologies has been accompanied by a rising public concern about potential adverse effects to human health, recently fueled again by observations of an increased incidence of cancer and DNA damage in exposed rats [1,2,3]. While EMFs used in wireless communication devices (wEMF) have been reported to affect various aspects of cell physiology, including the integrity of the genetic material [4,5,6,7,8], the underlying experimental evidence has been a subject of controversy both in the scientific community and in the public. Indeed, the exposure-related effects are often at the borderline of methodological sensitivity and statistical significance and, therefore, challenging to reproduce in independent settings [8,9]. Nevertheless, evidence for an impact of wEMFs on genome integrity should be carefully investigated, as this may indicate a mutagenic potential, which, even if minor, may be associated with an increased risk of cancer or degenerative human diseases.

The impact of environmental agents on DNA integrity can be assessed in different ways. Conventional experimental approaches include methods like the single cell gel electrophoresis (also called Comet assay (CA)), the detection of sister chromatid exchanges (SCE), and the quantification of micronuclei (MN) or nuclear DNA repair foci (γH2AX, 53BP1) [10,11,12]. These assays generally provide a sensitive readout with a reproducible dose-effect correlation and have also been applied for genotoxicity screening of non-ionizing EMFs. In some cases, these assays yielded unexpected positive results, implicating a DNA-damaging or damage-accumulating effect of RF-EMFs. A DNA-damaging effect of RF-EMFs, however, is difficult to explain on the basis of current knowledge; unlike ultraviolet radiation or gamma rays, the energetic impact of non-ionizing RF-EMFs is orders of magnitude too low to break or modify the chemical structure of DNA [13]. In the absence of a plausible mechanism, it has been proposed that effects of EMF on DNA integrity may reflect an indirect readout for changes in the cellular physiology, such as an enhanced formation and/or accumulation of reactive oxygen species (ROS) or an impairment of DNA damage checkpoints or maintenance mechanisms [4,7]. Likewise, we reasoned that the slightly increased levels of DNA damage, reported for RF-EMF-exposed cells in some studies, might originate from an accumulation of physiologically generated but unrepaired DNA damage, from replication intermediates, or from altered cellular ROS homeostasis. All these possibilities implicate an involvement of the DNA base excision repair (BER) or single-strand break repair (SSBR) system, as these DNA repair activities deal with the vast majority of physiological DNA damage, induced by cellular metabolism or by ROS [14,15,16,17]. XRCC1 is a central component in both BER and SSBR and functions as a scaffold and matchmaker protein. It forms distinct repair foci in the nucleus upon induction of DNA damage, indicating ongoing repair [18,19]. Given the important role of BER and SSBR in DNA repair, we reasoned that monitoring and/or modulating the dynamics of these processes provides a sensible approach to test a putative interference of wEMFs with the processing of physiological DNA damage.

The obvious controversy on the genotoxicity of RF-EMFs in general and wEMFs in particular prompted us to revisit the question in a systematic and also explorative manner in a joint effort of two laboratories with expertise in genetic toxicology and DNA repair. Here, we present the results of classical (Comet and sister chromatid exchange assays) and novel in vitro approaches (live cell imaging) to assess the genotoxic potential of the wEMF modulations: Universal Mobile Telecommunications System (UMTS), Global System for Mobile Communications (GSM), wireless networking (WiFi), and Radio-Frequency Identification (RFID). Key experiments were independently performed in the two laboratories in Hannover (Germany) and in Basel (Switzerland), using identical cell lines, standardized experimental protocols, and the same exposure equipment. Moreover, to address the issue of reproducibility of published data, we evaluated the impact of methodological variables, such as cell culture conditions and methods for CA analysis, on the performance and robustness of the CA in the detection of low-level DNA damage [8,9,20]. To raise the sensitivity of the assays and to explore modes of interaction between wEMFs and cellular DNA metabolism, we applied an enzyme-modified CA and performed exposures under conditions of saturating and/or inhibited cellular DNA repair capacity. Finally, we applied live cell imaging technology to investigate wEMF-induced alterations in DNA repair dynamics.

## 2. Material and Methods

Detailed description of all experimental protocols are provided in the Appendix A section.

### 2.1. Cell Models and Culture Procedures

All cells were cultured under a controlled atmosphere (37 °C, 5% CO_2_, 95% humidity), according to the providers’ recommendations. We note that standardized cell growth and propagation protocols were applied in both laboratories, located at the Fraunhofer Institute for Toxicology and Experimental Medicine ITEM (Hannover, Germany; Lab 1) and the University of Basel (Basel, Switzerland; Lab 2). For details see Appendix A. Primary human MRC-5 lung fibroblast (25 population doublings) and human osteosarcoma (U-2 OS) cells were obtained from ECACC. The immortalized human trophoblast cell line, HTR-8/SVneo, was generated by Graham [21] and provided by Dr. Elena Fabbri [22]. To obtain GFP-tagged XRCC1 cells, the coding sequence of XRCC1 (Genebank refseq NP_006288.2) was cloned in pEGFP-N1 (Invitrogen, Carlsbad, CA, USA) and the plasmid was transfected into U-2 OS cells. Stable clones were selected and characterized for adequate expression levels of the full-length fusion protein.

### 2.2. Exposure Equipment and wEMF Signal Modulations

The radio-frequency EMF exposure systems were built and provided by the Foundation for Information Technologies in Society (IT’IS foundation; Zurich, Switzerland) and are described in detail on https://itis.swiss/customized-research/emf-exposure-systems/in-vitro-sxc/. The sXc1950 setup is based on two R18 waveguides (exposed and sham-exposed), operating at 1950 MHz, and placed inside commercial incubators to ensure constant culture conditions. The EMF level in the waveguide was monitored by a detector diode, coupled into the waveguide and the RF generator output controller to maintain the target field strength. Calibration was achieved with probes, traceable back to national standards. Specific absorption rate (SAR) level and distribution was simulated and confirmed by measurements; SAR homogeneity was found to be better than 30%. The miniaturized sXcLive2450 system operates at 2450 MHz and comprises a dual mode TE102 cylindrical cavity with quadrature feed, which can be integrated into standard microscopy tables, providing 10% SAR homogeneity over a 6 mm diameter area, when used with a 100× objective (Appendix A, Appendix A). Computer-controlled signal and monitoring units were developed (1) to generate complex wEMF signals, (2) to continuously monitor the field and environmental conditions, and (3) to realize blinded exposure protocols. Cells were intermittently (5/10 min on/off) or continuously (for genotoxicity and live cell imaging, respectively) exposed to wEMF for the indicated periods of time and intensities. Signals were either without signal modulation (carrier wave) or with generic wEMF modulations for GSM [22,23], UMTS (Appendix A), WiFi (Appendix A, Appendix A), or RFID (Appendix A, Appendix A), based on these sources [24,25,26,27,28,29]. For details, see Appendix A. All wEMF exposure experiments were performed blinded for the experimenter. Decoding and quality control were done by the IT’IS foundation after finalization of data analysis.

### 2.3. Assessment of Genotoxicity

Alkaline Comet assays were done according to Singh et al. [30] with some modifications [31,32,33] (for details, see Appendix A). In brief, cells were carefully mixed with freshly prepared low-melting agarose, spread on slides pre-coated with agarose, and covered with a cover slip until solidification. After incubation of slides in lysis buffer and additional enzymatic treatment the human 8-oxoguanine DNA Glycosylase (hOGG1), where indicated, cellular DNA was unwound in alkaline buffer (pH > 13), and electrophoresis was performed. Nuclear DNA was stained, and DNA damage was analyzed either by Metafer CometScan (MetaSystem, Altlussheim, Germany) or by the Comet Assay III software (Perceptive Instruments, Haverhill, UK). Routinely, at least 100 nuclei were analyzed per slide. The parameter “% DNA in tail” (also called “tail intensity”) was chosen as primary readout as it was shown to be linear over a wide range and can be compared between different analysis systems. Single cell data of each slide/experimental condition were then summarized as arithmetic means, and, finally, the arithmetic means of at least three independent exposure experiments were statistically analyzed by 1- or 2-way ANOVA with post-hoc pairwise comparison tests in GraphPad Prism 8 and/or by paired and unpaired Student’s *t*-tests in MS Excel or SigmaStat software (Systat Software GmbH, Erkrath, Germany), considering *p* ≤ 0.05 as statistically significant.

For sister chromatid exchange (SCE) assays as described previously [34], the DNA of HTR-8/SVneo cells was labelled by culturing the cells in medium supplemented with 10 µM bromodeoxyuridine (BrdU) for 64 h. wEMF exposure and/or treatment with 2 µM of the PARP inhibitor AG-014699 (Selleck Chemicals, Houston, TX, USA) was done from 24–48 h, followed by 16 h of recovery. Differential staining of chromatids of metaphase spread chromosomes was done by Giemsa staining upon UV-B treatment of Hoechst 33258-labelled chromatin. Blinded for the examiner, the number of SCE, break points and chromosomes per cell were counted. Pooled SCE data and arithmetic means of the indicated number of independent experimental replica were statistically analyzed by ANOVA and Student’s *t*-test, respectively, using the GraphPad PRISM 8 software package.

### 2.4. Live Cell Imaging of XRCC1 Recruitment

The exposure chamber sXcli-2450 (https://itis.swiss/customized-research/emf-exposure-systems/in-vitro-sxc/sxclive2450/) for live-cell imaging was mounted on a Leica 6000B epifluorescence microscope, equipped with a temperature-controlled incubation chamber and an UV-A laser at 355 nM (Q-switched CristaLaser). To induce DNA damage, defined nuclear regions were irradiated with the UV-A laser [35] and recruitment of XRCC1-GFP fusion protein was imaged sequentially in the indicated time intervals. To avoid any experimenter bias, sham or wEMF exposures were performed blinded for the observer, and all image processing and quantitation was done in an automated way, using the open source software “CellProfiler” (http://www.cellprofiler.org) [36]. Relative XRCC1 recruitment in the indicated number of assessed nuclei from several independent experiments was statistically analyzed by two-way ANOVA for repeated measurements and post-hoc Dunnett’s multiple comparisons test of matched time-points using the GraphPad PRISM 8 software package.

## 3. Results

### 3.1. Replication Experiments on Genotoxic Effects of GSM-Modulated Signals

Over the years, there has been a number of studies addressing the genotoxic potential of RF-EMFs. Two studies in particular indicated that exposure to generic GSM signals (SAR 2 W/kg) can increase the level of DNA damage in primary human fibroblasts and in immortalized trophoblast cells [22,23]. We took these observations as starting point to investigate the genotoxic potential of wEMF exposure in more detail. To validate the published effects on trophoblasts [22], we first assessed DNA damage in HTR-8/SVneo trophoblasts, exposed to an intermittent (5/10 min on/off) GSM modulation for 1, 4, and 24 h. In alkaline CAs, we did not detect differences in DNA damage levels between sham- and GSM-217Hz-exposed (2 W/kg SAR) cells. These experiments were performed independently in two laboratories with consistently negative results (Figure 1a,b). Performance of the CA method was judged to be adequate, as induction of DNA damage was readily detected for both positive controls, 1 Gy of γ-ray and the known clastogen ethyl methanesulfonate (EMS) (Figure 1c). Similarly, we were unable to reproduce the finding that a GSM signal as well as an unmodulated carrier wave (continuous wave) were able to increase DNA strand breaks in primary human fibroblasts [23]. We found no significant DNA-damaging effect in CAs, when intermittently (5/10 min on/off) exposing primary human MRC-5 fibroblasts to 2 W/kg SAR GSM-modulated signals (Figure 1d) or to the 1.95 GHz carrier wave (Figure 1e) in an exposure dose (0, 0.5, 2, and 4.9 W/kg SAR) and exposure time-dependent (1, 4, and 24 h) manner (Appendix A).

Reproducibility can fail for many reasons. To evaluate the published results, we did our best to replicate the exact experimental design, CA scoring, and evaluation procedures, which led to the originally reported findings [22,23]. We performed a series of extra experiments, in which we quantified DNA damage in parallel by visual (as in Diem et al. [23]) and automated scoring of CAs, using the originally used ES-1 primary human fibroblast cell line. Yet, we were not able to measure increased DNA damage following exposure of ES-1 cells to an intermittent (5/10 min on/off) 1950 MHz GSM-talk modulated signal at SAR values of 1 and 2 W/kg for 16 h, irrespective of visual or automated CA analysis (Appendix A). We also tested genotoxicity of GSM in the primary fibroblast line HR-1d that showed higher sensitivity in CAs than ES-1 and MRC-5 cells in a 50 Hz MF exposure setting [37]. By visual scoring, we observed a small, but reproducible increase of DNA damage after exposure of HR-1d cells to an intermittent (5/10 min on/off) GSM-talk modulated signal at SAR values of 1 and 2 W/kg for 16 h (Appendix A). Automated CA scoring, however, produced inconsistent results, showing a small effect at 1 W/kg SAR but none at 2 W/kg (Appendix A). Notably, exposure of HR-1d cells to the unmodulated 1950 MHz carrier wave (1 W/kg SAR) produced no consistent CA effect, irrespective of the scoring method (Appendix A). Based on these results, we conclude that exposure of primary human fibroblasts to GSM-modulated signals does not induce detectable DNA damage in a reproducible manner in CAs, although trends were evident under specific experimental conditions.

Based on these observations, we reasoned that variations in CA methodology, including data analysis, are a possible source of incongruence in the assessment of “small effects”, and may explain the contradictory CA findings in literature. To address this assumption, we tested the performance of visual versus automated CA scoring more systematically. While the visual CA scoring can be criticized for its non-continuous, cell classification-based estimation of DNA damage, which largely depends on the judgement of the evaluator [38], automated methods are more objective but have the disadvantage of a limited “intelligence” in interpreting unexpected events. We, therefore, applied visual and automated scoring to a defined CA dataset, generated by treating MRC-5 cells with a low dose of the DNA-oxidizing agent H_2_O_2_ (10 µM) for 15 min. Although the numerical outputs (% DNA in tail) differed slightly, the low level of induced DNA damage was detectable with all four scoring methods applied (visual scoring, semi-automated scoring by the CometScore or Comet Assay IV software or fully automated analysis by the Metafer CometScan system) (Appendix A). The quantitation of DNA damage of individual nuclei varied with the underlying image quantitation algorithms, but there was a good correlation between the quantitation methods, regarding the population-based assessment of DNA damage (Appendix A). We then compared the CA scoring methods in terms of reproducibility and sensitivity, by analyzing MRC-5 cells treated with a range of low concentrations of H_2_O_2_ (0–70 µM) for 10 min. The analysis showed that visual scoring was capable of robustly picking up low-level DNA damage, induced in the presence of 30 µM H_2_O_2_, while the automated methods required concentrations of ≥50 µM H_2_O_2_ to reproducibly indicate DNA damage (Appendix A). Such small differences in method performance may in part explain the discrepancies in the measurement of low-level effects. An additional source of effect variation might be the culturing conditions and/or adaptation of cells. To address the impact of cell culture conditions on reproducibility of CA readouts, we revisited our former observation of an increase of the Comet tail factor (visual scoring) following exposure of MRC-5 cells to a 50 Hz ELF-MF [37]. In line with previous results, automated CA analysis detected a small increase in DNA damage following 50 Hz ELF-MF exposure in early passage MRC5 cells. This effect, however, disappeared, when cells were exposed to 50 Hz ELF-MF in a later passage (Appendix A). This clear impact of cell passage or cell state on DNA damage prompted us to repeat the exposure of HTR-8/SV40neo trophoblasts to a GSM-modulated signal for 24 h (GSM-217Hz, 5/10 min on/off, 2.2 W/kg SAR) with cells of two different sources and under serum-free conditions during exposure as done by Franzellitti et al. [22]. However, the variation of cell culture conditions did not yield to increased DNA damage in the CA experiments, as reported previously (Appendix A).

### 3.2. Investigation of the Genotoxic Potential of UMTS-, WiFi-, and RFID-Modulated wEMF

To address the genotoxic potential of other relevant wEMFs, we exposed MRC-5 and HTR-8/SVneo cells to generic UMTS, WiFi, and RFID modulations at different exposure doses (0, 0.5, 2, 4.9 W/kg SAR) and durations (1, 4, and 24 h) (Appendix A). We performed both conventional alkaline as well as hOGG1-modified (oxidative) CAs to enhance the sensitivity of our analyses and to get information regarding mechanisms of action. The most extreme wEMF exposure conditions (highest SAR, longest exposure duration) were again assessed independently in the two laboratories (Figure 2, compare left and right panels). Compared to the background DNA damage levels, no changes were detected in MRC-5 cells, intermittently (5/10 min on/off) exposed to 4.9 W/kg UMTS-, WiFi-, or RFID-modulated signals for 24 h (Figure 2a). Likewise, the exposure of HTR-8/SVneo trophoblasts to the UMTS and WiFi signals at a SAR of 4.9 W/kg did not result in significantly altered CA readouts (Figure 2b).

Exposure to wEMFs has been proposed to alter levels of intracellular reactive oxygen species (ROS) [4,5,6,7], which in turn may damage DNA. Elevated nuclear ROS or lipid peroxidation products frequently result in oxidation of guanidine bases to form 8-oxo-G [39]. In humans, this common pre-mutagenic DNA lesion is predominantly recognized and eliminated by the DNA repair enzyme hOGG1 to avoid excessive genetic mutations. Considering that wEMF exposure might induce low levels of oxidative DNA base damage, we reasoned that there might be a higher chance to detect those lesions by a variant of the alkaline CA that involves enzymatic DNA pre-processing with hOGG1 prior to gel-electrophoresis [33]. Yet, the assessment of oxidative DNA damage in MRC-5 and HTR-8/SV40neo cells did not reveal consistent DNA-damaging effects after exposure to UMTS-, GSM-, WiFi-, or RFID-modulated signals or the respective carrier wave at different doses and for different durations (Figure 3, Appendix A), while incubation with KBrO_3_, as the positive control for oxidative DNA damage, resulted in reproducibly increased CA readouts. Overall, we conclude that there is no indication for a direct or ROS-mediated DNA-damaging potential of all tested wEMF signals under the experimental procedures and conditions applied.

### 3.3. Modulation of DNA Repair Capacity does not Reveal wEMF-Dependent DNA Damage

There is currently neither conclusive evidence nor a plausible mechanism for a direct damaging interaction of wEMFs with DNA. Nevertheless, CAs have indicated slight DNA-damaging effects, which need to be explained. One possible scenario is an indirect action such as an interference of wEMFs with cellular DNA repair that results in an accumulation of spontaneously occurring DNA damage [40,41]. Additional cell stressors, including unfavorable cell culture conditions, may contribute and potentiate such indirect wEMF effects. We, therefore, reasoned that compromising the DNA repair systems, which eliminates the bulk of continuously occurring DNA base damage in cells, i.e., DNA base excision repair (BER) and single-strand break repair (SSBR), may amplify such wEMF effects and uncover DNA repair intermediates (i.e., DNA strand breaks), when analyzed by CAs. Pharmacological inhibition of poly-ADP ribose polymerase 1/2 proteins (PARP1/2) represents an opportunity to interfere with the BER and SSBR process at early stages [42]. We, therefore, assessed the efficacy of the PARP inhibition approach in the cell lines used in this study by measuring H_2_O_2_-induced poly-ADP ribose (PAR) synthesis. In cell cultures, titrating the PARP1/2 inhibitor (PARPi) AG-014699 resulted in a significant reduction of PAR synthesis following H_2_O_2_ treatment, both in MRC-5 and in HTR-8/SVneo cells (Appendix A). In line with an involvement of PAPR1 in the repair of oxidative DNA damage, CA analyses revealed delayed removal of DNA damage after H_2_O_2_ exposure for 30 min in PARPi-treated MRC-5 and HTR-8/SVneo cell populations (Appendix A). Thus, PARP inhibition delays the repair of DNA base damage in these cell systems and thereby leads to the generation of CA-detectable abasic sites and DNA strand breaks following DNA damage induction.

We then assessed the level of DNA damage in MRC-5 (UMTS, GSM, WiFi; Figure 4a) and HTR-8/SVneo (UMTS, GSM, WiFi, RFID; Figure 4b) cells after intermittent wEMF exposure (5/10 min on/off, 4.9 W/kg) for up to 24 h in the presence or absence of 2 µM PARPi. Although tendencies were notable, PARP inhibition did not consistently alter DNA damage levels in cells exposed to wEMF, showing that limiting BER capacity does not synergize with wEMF effects in the accumulation of DNA damage. In PARPi-treated MRC-5 cells, exposed to the UMTS signal modulation at 4.9 W/kg SAR, however, we observed differences in DNA damage level between sham and UMTS exposure for 16 h (*p* = 0.079) and 24 h (*p* = 0.075). These differences, however, were inconsistent, when comparing UMTS exposures for 16 and 24 h (reduced and increased % DNA in tail, respectively).

To investigate this observation in more detail, we combined UMTS exposure of MRC-5 cells with a DNA-damaging treatment that challenges the DNA repair capacity. We used the DNA alkylating agent EMS at a low dose (0.25 µL/mL, 1 h prior to CA) that approximately doubled the background level of detectable DNA damage in CAs (Figure 4c). Similar to PARP inhibition, the co-exposure with EMS revealed an indication for a transient accumulation of DNA damage by intermittent UMTS exposure at 4.9 W/kg SAR. This effect was significant for 1 h, still notable for 4 h, but lost for 24 h of UMTS exposure (Figure 4c, Appendix A), suggesting that the UMTS signal may elicit a temporary stress response in MRC-5 cells, which impairs their response to EMS, rather than inducing DNA damage itself or compromising DNA repair capacity.

### 3.4. Dynamics of DNA Repair is not Affected by UMTS Exposure

To address whether wEMF exposure may alter the dynamics of DNA repair, we investigated the efficiency of the cellular DNA repair machinery to locate and associate with DNA damage. XRCC1 is a central player in BER and SSBR and functions as a scaffold protein that coordinates the individual repair steps. The engagement of XRCC1 in DNA repair can be visualized owing to its property to form distinct repair foci in the cell nucleus [18,19]. We reasoned that monitoring the dynamic behavior of XRCC1 foci at sites of DNA damage will provide a relevant readout of a potential interference of wEMF with DNA repair. To facilitate such an analysis, we developed the sXcLive2450 exposure device that is integrated into a microscope system, allowing for live cell imaging during wEMF exposure. Making use of the sXcLive2450 device, we monitored the recruitment of GFP-tagged XRCC1 (ectopically expressed in U-2 OS cells) to sites of micro-laser induced oxidative DNA damage under sham or UMTS exposure conditions. Blinded to the examiner, cells were pre-exposed to either a UMTS signal (2 W/kg) or sham-exposed for 15 min before local induction of DNA damage by a UV-A micro-laser and subsequent imaging under continued UMTS exposure for 15 min (Figure 5a). Sham-exposed cells showed a fast focal accumulation of XRCC1, peaking at about 1 min after DNA damage induction, followed by a gradual dissociation over time (Figure 5b). In these experiments, UMTS exposure did not significantly alter the XRCC1 recruitment kinetics. By contrast and as expected [42], PARP inhibition (5 µM AG-014699) for 2 h prior to DNA damage induction significantly reduced XRCC1 recruitment in such a way that no peak of enrichment was discernible anymore.

To take a closer look at the early, rapid phase of DNA damage recognition and XRCC1 recruitment, we also imaged in 5 s intervals for up to 4 min. Cells were pre-exposed to a continuous UMTS signal (2 W/kg) for 15 min before local induction of DNA damage. In this experiment, we induced lower levels of DNA damage (only about 10% of the total GFP-tagged XRCC1 was recruited to the site of laser-induced damage) to avoid saturation effects that might mask subtle impacts of the exposure on early repair events. We observed a rapid accumulation of XRCC1 within 30 s following DNA damage induction, and then further recruitment at a lower rate peaking at about 3 min after damage induction in sham-exposed cells (Figure 5c). Although a slightly lower initial rate of XRCC1 recruitment was discernible in UMTS-exposed cells, this difference did not reach statistical significance. Control cells, pre-treated with PARP inhibitor (5 µM AG-014699) for 30 min, however, showed a significantly reduced XRCC1 recruitment 30 s after damage induction. Overall, we conclude that the dynamics of central steps of BER and SSBR processes are not significantly altered by UMTS exposure.

### 3.5. Sister Chromatid Exchange is Unaffected in Cells Exposed to UMTS

Up to this point, our data suggest that wEMF exposure neither induces direct DNA damage nor significantly alter DNA repair capacity and dynamics in human cells. Yet, cells exposed to a UMTS signal provided some indication for slightly higher DNA damage in CA when co-treated with a PARPi or EMS. We reasoned that perturbations caused by these co-exposures might lead to a small increase in the steady-state of unprocessed spontaneous DNA base damage and single-strand breaks, which during DNA replication will trigger alternative repair by homologous replication (HR), involving the sister chromatids [43,44,45]. Repair by HR between sister chromatids often results in a reciprocal exchange of sequences that can be visualized in the sister chromatid exchange (SCE) assay. In line with this mechanism, the treatment of HTR-8/SVneo cells with the PARPi AG-014699 resulted in a 4–5-fold induction of SCE events and chromosomal break points (BP) (Figure 6). Intermittent exposure (5/10 min on/off) to a UMTS signal for 24 h (4.9 W/kg), however, did not cause any notable change in SCE events, irrespective of the presence or absence of PARPi (Figure 6b). This said, it is known that the chromatin-labelling process by BrdU itself has a recombinogenic potential. We used two different batches of BrdU, each in two experiments, exhibiting differences in basal SCE levels by a factor of two (Appendix A). When comparing sham- and UMTS-exposed cells within BrdU batches, a small but consistent increase in exposed cells becomes apparent in all experiments without PARP inhibitor, yielding a significant difference in paired statistics or upon normalization to sham exposure (Appendix A).

## 4. Discussion

Recently, indications for a higher incidence of certain types of cancer upon lifelong wEMF exposure of animals [1,2] have reactivated a long-lasting dispute about if, and how, wEMF exposure may induce tumorigenesis [46]. Interactions of wEMFs with various cellular targets and processes have been reported, also including effects on genome integrity in human cells [5,7,8,20]. Classical and well-established assays for genotoxicity assessment have been used with overall inconsistent and, hence, inconclusive results. The discrepant findings prompted us to systematically assess the genotoxic potential of four relevant wEMF signal modulations (UMTS, GSM, WiFi, and RFID) in cultured human cells. We intended to replicate key aspects of previous studies, to elucidate possible causes of discrepancies and, furthermore, to advance the mechanistic insight into potential interactions between wEMF exposure and the DNA. All experiments were conducted in a blinded manner to exclude any experimenter bias, and key experiments were assessed independently with identical protocols in two laboratories. This approach was previously practiced in a genotoxicity screening study with human blood cells after exposure to GSM-modulated signals [47].

Prompted by the findings of others [22,23], we decided to focus our work on established cell models, i.e., human primary fibroblast and immortalized trophoblast cells, previously used for the investigation on the genotoxic potential of EMFs. First, we assessed the genotoxic potential of wEMF signals as well as of the unmodulated 1950 MHz continuous carrier wave, making use of the alkaline CA (Figure 1 and Figure 2; Appendix A). We tested more than 40 wEMF exposure conditions, none of which produced statistically significant alterations in the level of CA-detectable DNA damage. The results, cross-validated in two laboratories, thus demonstrate that the exposure of cultured human fibroblast and trophoblast cells to relevant wEMF signal modulations does not induce alkaline CA-detectable DNA damage. This is in contrast to previously reported CA effects, using comparable or even identical exposure conditions and/or cell lines [22,23,48], but is consistent with findings of other studies with similar experimental designs [40,41,49,50,51,52,53,54,55,56,57]. We did, however, observe a weak compound effect of UMTS exposure and EMS treatment (Figure 4c), detectable transiently at early time points of exposure but disappearing after longer exposure. At this point, we do not have an explanation for the transient nature of the effect, other than that UMTS exposure may generate a temporal cellular response that impairs DNA maintenance activities in a way that affects steady state levels of physiological DNA damage in cells. If so, an additional perturbation of cellular DNA repair capacity by EMS might reveal more DNA damage in wEMF-exposed cells in CAs.

Genotoxicity assessment under perturbation or DNA repair has been shown to improve the sensitivity and specificity of assays (e.g., chicken DT40 cell system [58]) and has been applied to address wEMF effects [40,41]. For instance, Sun et al. [41] sensitized immortalized murine fibroblasts by depleting ATM, a key factor in DNA damage response. They measured a transient increase in DNA strand break induction in the alkaline CA in wild-type cells, exposed for 1 h to a GSM-modulated signal at 4 W/kg SAR. In ATM-depleted cells this accumulation was shifted to the 12 h time-point. This effect correlated well with absolute levels as well as the proportion of phosphorylated XRCC1 in the cells, thus indicating an involvement of BER or SSBR. Here, we neither observed a disturbance of XRCC1 recruitment to sites of active BER in cells exposed to a UMTS signal (Figure 5) nor an increase in DNA strand breaks and oxidative DNA damage after exposure to GSM-modulated signals (Figure 1 and Figure 3; Appendix A).

Similar to our approach to reduce BER and SSBR capacity by PARP1/2 inhibition, Wang and colleagues [40] depleted the DNA repair enzyme OGG1 to sensitize murine cells and thereby identified OGG1 as a main player in the generation of GSM-induced DNA strand breaks. This observation implicated the occurrence of oxidative DNA damage, following exposure of murine cancer cells of neuronal origin to GSM-modulated signals for 24 h (SAR 2 W/kg). These effects were observed in an enzyme-modified CA, using the FPG protein to eliminate oxidized bases (mostly 8-oxo-G) from the DNA and to make them detectable by CA. Under similar exposure conditions but using hOGG1 instead of FPG in the enzyme-modified CA, we found no increase in oxidative DNA damage at even higher SAR levels in human cells, neither for GSM nor for other wEMF signals (Figure 3, Appendix A). However, the substrate spectrum of FPG exceeds 8-oxo-G lesions and we used primary human cell lines rather than mouse cancer cells in our study. Therefore, we cannot exclude cell type- and perhaps also tumor cell-specific factors such as a generally higher sensitivity of fast-dividing cancer cells towards environmental insults to play a role in these diverging outcomes.

It was previously reported that RF-EMF exposure (continuous carrier wave and UMTS signal) results in an induction of PARP1 in rat cells and that PARP inhibition can modulate the response to co-genotoxic treatment [59,60]. Our wEMF exposures under conditions of PARP1/2 inhibition failed to provide evidence for an interaction between BER and/or SSBR and wEMF exposure, both in primary MRC-5 lung fibroblast and HTR-8/SVneo cells. However, exposure of PARPi-treated cells with a UMTS signal for 24 h brought forward some hints for an accumulation of DNA damage (Figure 4).

Overall, our data do not point to a negative impact of wEMFs exposure on the general DNA repair capacity of human cells. However, we cannot fully exclude the possibility that wEMFs affect specific repair processes under specific conditions, as reported for instance for nucleotide excision repair in cancerous p53 negative cells [56]. Besides directly damaging DNA or perturbing DNA repair capacity, wEMFs might trigger a general stress response in cells, which then modulates cell behavior in the presence of additional stressors such as genotoxic agents or unfavorable cell culture conditions [61]. If that was the case, it is not surprising that the complexity of the interactions of environmental factors, combined with the variable genetic make-up and physiological state of cells, will generate cell type-specific [48,54] and difficult to reproduce experimental outcomes [6,20]. Cellular states would for instance also regulate the expression of factors involved in processes of the DNA metabolism among which replication is a vulnerable process with regard to external disturbance. Disruption of replication mediates specific genotoxic effects like an induction of micronuclei and enhancement of SCE frequency. We indeed noticed a tendency towards increased SCE formation in UMTS- compared to sham-exposed cells (Figure 6 and Appendix A). On the basis of the above reasoning, we would argue that this effect most likely results from a slight cell stress-induced perturbance of DNA replication rather than from a direct induction of DNA damage, for which we were not able to find any evidence. In line with this, we have previously shown that primary human fibroblast cells only responded to a 50 Hz MF, while actively dividing [37].

Finally, the reason for the discrepant findings in genotoxicity testing of wEMF exposure remains speculative. One may argue that our failure to reproduce the induction of DNA damage, previously observed in primary human fibroblasts [23,48], was due to the use of different cell lines with different origin under different experimental conditions. Yet, we tried to replicate the published conditions as precisely as possible and explored the potential role of additional biological (cell lines, DNA repair interference, co-exposure, and others) and experimental (culture conditions, CA data scoring, and others) variables, none of which yielded clear and reproducible evidence for an effect of wEMFs on DNA integrity in classical and adapted genotoxicity and DNA repair assays. We noted, however, differences in the sensitivity of visual and automated CA scoring methods, which may lead to an algorithm-dependent systematic bias in the estimation of DNA damage (Appendix A), and could be one explanation for the divergent observations on low-level of DNA damage induction such as reported for EMF exposure. For instance, there is still no consensus on how CA data should be appropriately handled with regard to statistics [62,63]. We, therefore, conclude that the CA, designed to measure DNA damage on the single cell level, may have limitations in monitoring cellular wEMF responses and produce weak positive results in one setup and a negative result in another, depending on the experimental conditions and applied method of analysis and statistic [38,47,64,65]. In our opinion, the cell culture method appears very critical for the determination of the genotoxic potential of wEMF exposure, as cellular stress might be induced to a variable extent, which modulates wEMF-dependent responses [61]. For instance, we were not able to measure increased DNA damage in GSM-217Hz-exposed HTR-8/SVneo cells, using complete standard growth medium (Figure 1 and Figure 4), whereas a weak tendency (Appendix A) and significantly increased DNA damage [22] were apparent, when the same cells were grown without serum at high cell density.

## 5. Conclusions

We investigated the genotoxic potential of modulated RF-EMF as used in wireless technology (UMTS, GSM, WiFi, and RFID) on cultured human cells. Classical and advanced genotoxicity testing and DNA repair assessment produced no conclusive evidence for a disturbance of DNA integrity or changes in the DNA repair capacity, following wEMF exposure. These overall negative results are in contrast to some previously reported positive findings. Investigating the underlying reasons for this discrepancy, we identified cell culture conditions and the CA methodology as likely relevant variables. In some experiments with UMTS exposure, we noticed small tendencies for wEMF exposure-associated changes in DNA damage levels and repair dynamics. In the absence of evidence for a direct DNA-damaging potential of wEMF, we interpret these to possibly be caused by an unspecific wEMF-induced cellular stress response. The nature of such an interaction between wEMF and cellular physiology, however, remains unclear and needs to be further investigated.

## Figures and Tables

**Figure 1 genes-11-00347-f001:**
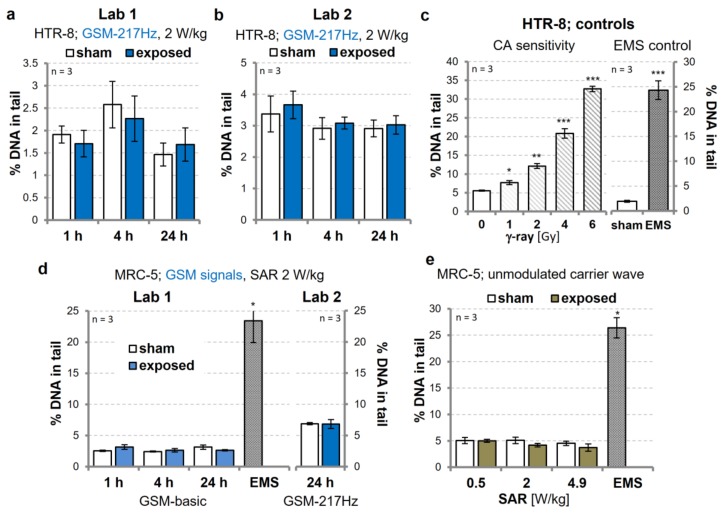
Global System for Mobile Communications (GSM)-modulated signals do not induce DNA damage in human cells. Immortalized HTR-8/SVneo (HTR-8) trophoblasts (**a**–**c**) and primary MRC-5 lung fibroblasts (**d**,**e**) were intermittently exposed (5/10 min on/off) to GSM-217Hz (**a**,**b**,**d** right panel), a GSM-basic signal (**d** left panel), or an unmodulated RF-EMF (**e**, carrier wave), as indicated. Alkaline Comet assays (CAs) were independently executed in two laboratories, according to their standard protocols, and compared to sham-exposed control samples. (**c**) As sensitivity and positive controls, cells were irradiated with increasing doses of γ-ray or treated with 0.5 µL/mL of ethyl methanesulfonate (EMS) for 1 h, concomitant to RF-EMF exposure. Data represent arithmetic means ± SEM of three independent experiments. Asterisks indicate significance levels of Student’s *t*-test comparing GSM-exposed and EMS-treated cells with sham-exposed samples: *****
*p* < 0.05; ******
*p* < 0.01; *******
*p* < 0.001.

**Figure 2 genes-11-00347-f002:**
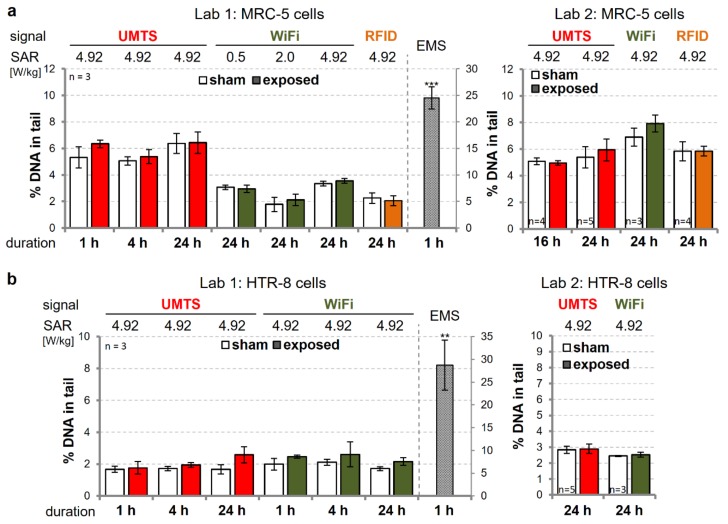
Genotoxicity assessment by CA of modulated electromagnetic fields relevant for wireless communication (wEMF) in human cells. Primary MRC-5 lung fibroblasts (**a**) and immortalized HTR-8/SVneo (HTR-8) trophoblasts (**b**) were exposed to indicated doses of intermittent (5/10 min on/off) wEMF signals (colored bars by signal type) or sham-exposed (white bars) for the duration indicated. Applying standardized experimental procedures, DNA damage was assessed by the alkaline CA, independently executed in two laboratories (compare left and right panels). As a positive control, cells were treated with 0.5 µL/mL ethyl methanesulfonate (EMS) for 1 h, concomitant to wEMF exposure. Data represent arithmetic means ± SEM of independent experiments (*n*). Asterisks indicate significance levels of Student’s *t*-test comparing wEMF-exposed and EMS-treated cells with sham-exposed samples: ******
*p* < 0.01; *******
*p* < 0.001.

**Figure 3 genes-11-00347-f003:**
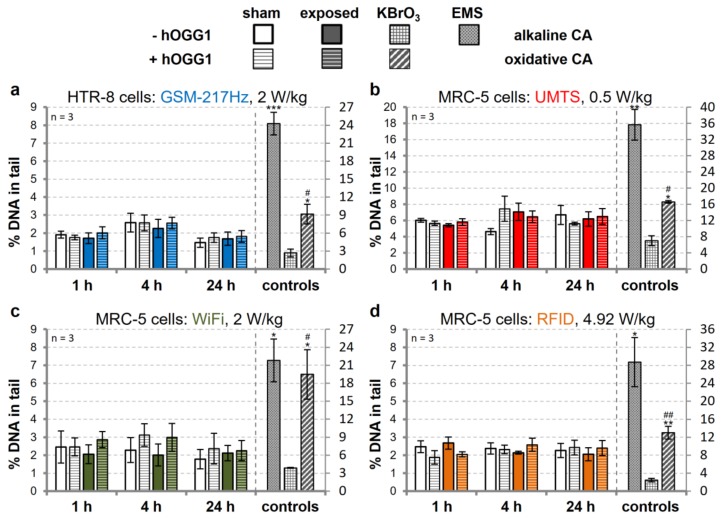
Assessment of oxidative DNA damage in wEMF-exposed human cells. Immortalized HTR-8/SVneo (HTR-8) trophoblasts (**a**) and primary MRC-5 lung fibroblasts (**b**–**d**) were exposed to the indicated doses of intermittent (5/10 min on/off) wEMF signals (colored bars by signal type) or sham-exposed (white bars) for the duration indicated. Concomitant to wEMF exposure, cells were treated with 0.5 µL/mL EMS (positive control for DNA strand break induction) and 0.5 mM KBrO_3_ (positive control for induction of oxidative DNA damage) for 1 and 4 h, respectively. Exposed/treated cells were divided and subjected to enzyme buffer- or hOGG1-incubation before electrophoresis. Data represent arithmetic means ± SEM of three independent experiments. Asterisks and hashes represent the statistical significance levels of Student’s *t*-test: *****^,#^
*p* < 0.05; ******^,##^
*p* < 0.01; *******
*p* < 0.001, comparing wEMF-exposed as well as control cells with sham-exposed samples and data of hOGG1-treated (oxidative DNA lesions) slides with those of buffer-treated slides (DNA strand breaks only), respectively.

**Figure 4 genes-11-00347-f004:**
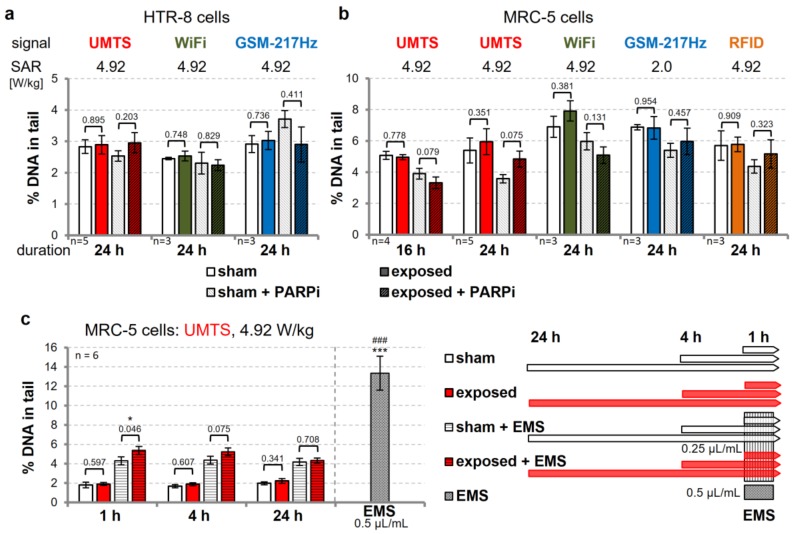
Reducing DNA base excision repair (BER)/single-strand break repair (SSBR) capacity and co-genotoxic treatment does not reveal DNA damage in wEMF-exposed cells. Human HTR-8/SVneo trophoblasts (**a**) and primary MRC-5 lung fibroblasts (**b**) were exposed to intermittent (5/10 min on/off) wEMF signals (colored bars by signal type) or sham-exposed (white bars) in the presence of 2 µM PARP1/2 inhibitor AG-014699, as indicated. (**c**) Primary MRC-5 lung fibroblasts were exposed to an intermittent (5/10 min on/off) Universal Mobile Telecommunications System (UMTS) signal (red bars) at 4.92 W/kg SAR or sham-exposed (white bars) for the duration indicated. During the last 1 h of exposure, half of the samples were treated with 0.25 µL/mL of ethyl methanesulfonate (EMS). To assess dose response, unexposed cells were incubated with 0.5 µL/mL EMS for 1 h. DNA damage levels were assessed by the alkaline CA. Shown are the arithmetic means ± SEM of the indicated number of independent experiments. The significance level (*p*-values) of paired Student´s *t*-tests are given, comparing sham- and wEMF-exposed cells. Asterisks and hashes represent the significance levels: *****
*p* ≤ 0.05, *******^,###^
*p* < 0.001, comparing the positive control EMS with the matched sham exposure and sham + EMS samples, respectively.

**Figure 5 genes-11-00347-f005:**
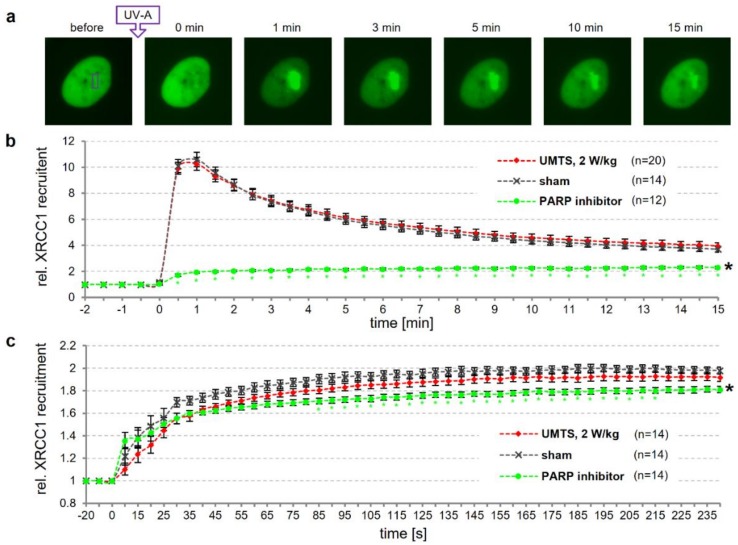
Dynamics of XRCC1 association with UV-A-induced DNA damage under sham or UMTS exposure. XRCC1-GFP expressing U-2 OS cells were sham- or pre-exposed with continuous 2 W/kg UMTS signal (t > 15 min) before induction of DNA damage by a UV-A laser in a pre-defined nuclear region (rectangle). Recruitment of XRCC1 to sites of DNA damage was recorded by live cell imaging. (**a**) Representative images of time-dependent XRCC1 recruitment. (**b**,**c**) Automated quantitation of XRCC1 recruitment (ratio median pixel intensities of DNA damage site and nucleus) to sites of high (**b**) and low (**c**) dose of DNA damage (duration of micro-laser irradiation) monitored for 15 and 4 min, respectively. As a control for altered BER/SSBR dynamics, cells were pre-incubated with 5 µM of the PARP inhibitor AG-014699. Error bars represent SEM of standardized XRCC1 signals of *n* = 12–20 analyzed nuclei, pooled from four independent experiments. Black and colored asterisks mark significant differences (*p* ≤ 0.05), compared to sham-exposed cells by two-way ANOVA and post-hoc Bonferroni test for matched time-points, respectively.

**Figure 6 genes-11-00347-f006:**
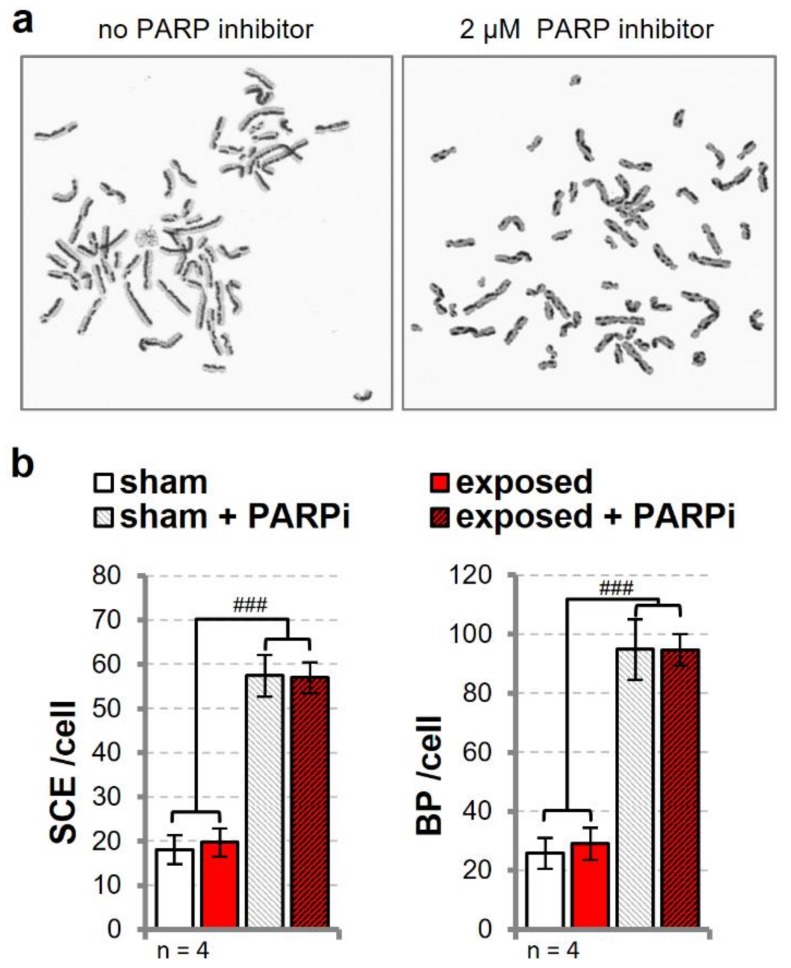
Assessment of sister chromatid exchanges (SCEs) in UMTS-exposed cells. Human trophoblast HTR-8/SVneo cells were cultured in the presence of 10 µM bromodeoxyuridine (BrdU) for two cell cycles. During the second cell cycle, cells were exposed to an intermittent (5/10 min on/off) UMTS signal (red bars) at 4.92 W/kg SAR or sham-exposed (white bars) for 24 h either in the presence or absence of 2 µM of the PARP inhibitor AG-014699. The number of SCEs and break points (BP) were counted in metaphase spreads. (**a**) Representative images of metaphase spreads with differentially labelled chromatids of HTR-8/SVneo cells proficient or impaired for BER, and (**b**) statistical analysis of four independent experiments by two-way ANOVA and post-hoc Tukey’s multiple comparisons test. Error bars indicate SEM. Asterisks and hashes represent the significance levels (^###^
*p* < 0.001) of pairwise comparison of matched samples for UMTS and PARP inhibitor effects, respectively.

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
