# Peer review of "Assessment of Genotoxicity in Human Cells Exposed to Modulated Electromagnetic Fields of Wireless Communication Devices"

_genes, 2020, doi:10.3390/genes11040347_

Round 1
Reviewer 1 Report
This paper reports on possible DNA breakage and repair in response to exposure to several forms of Radiofrequency (RF) radiation used in modern telecommunications. It concentrates on a particular aspect of the Comet Assay (%DNA in tail) and secondly the dynamics of repair after UV damage, with and without RF. By concentrating on a small number of end-points the study avoids the weakness of other recent studies on RF in living rodents of employing large numbers of comparisons, without adjustment.
The basic finding of no effects of RF on either DNA damage or rate of repair after UV damage is a highly important result and deserves to be published, particularly since the study uses several strategies to ensure quality, specifically: the collaboration between two laboratories; the use of well calibrated and understood exposure apparatus; the use of blinding in exposure conditions and scoring of results; the provision of raw data in supplementary material; the use of sham and positive controls; the use of more than one intensity of RF etc.
There are however a number of points of clarification and correction which need to be carried out before publication, as below:
1. P 9 lines 302 – 309. P values of > 0.05 have been declared as ‘Not significant’, so there seems to be no logic in discussing ‘trends’ with P ~ 0.08 for both time points. The phrase ‘nearly reaching statistical significance’ on line 307 seems also not to be justified. All that can be said is that p values are lower than most. This also appears on p 14, line 463 and p12 line 379 (and appears to be the reason for studying SCE in UMTS only). On the other hand, on p 10 line 327 the word ‘trend’ is used for a P value of 0.046, which is actually (just) under 0.05, so should score an asterisk (*) to be consistent with earlier figures. Some revision of language at these points is advised. Incidentally, S18 reports a P value of 0.042, which also qualifies for a *, but there is no comment on this. A more mundane criticism is that the order of panels b & c in Fig. 4 is reversed, perhaps in order to save space, but makes comprehension difficult. Perhaps the legend should draw attention to the change in order.
2. As a point of clarification in Fig 4c, is the only difference between ‘exposed + EMS’ and ‘EMS’ (control) that the latter has 0.5 rather than 0.25 mL/mL EMS? It appears that both have ‘concomitant .. UMTS exposure’. If this is so, then doubling the [EMS] seems to increase the effect by 11/3 or nearly 4 times. If this is a particularly sensitive region of EMS response then small discrepancies in EMS added could give rise to additional variability (and hence to the low P values mentioned).
3. Overall there seem to be two clusters of sham % DNA in tail around 2% and around 6%. This is particularly noticeable in Fig 2a. It is not a lab-specific thing, since both labs are reporting similar clusters. Some comment or explanation would be in order.
4. The types of RF used are apparently colour coded, although I could not find any mention of a key to the colours. However, it becomes apparent on careful reading that the colours do refer to different types of RF exposures. Would help to have a key.
5. Regarding the acronyms GSM, UMTS, WiFi, RFID (p 1 line 29 and elsewhere): these would not normally be familiar to readers of Genes and should be defined somewhere early on.
6. The description of the exposure system could be improved. Section S1.2 does give an overview and links to pages with further details. However, the precise method of ensuring that the SAR values are as stated in the main paper needs to be included (presumably via modelling). The deviations of SAR uniformity (given as <30% in S1.2) should also be mentioned in the main paper. The description of the microscope-based system is also less than optimal (the web page gives a very poor diagram of the actual system and no details of the derivation of SAR values). The phrase ‘excellent SAR homogeneity’ in S1.2 is unsupported hyperbole. The link from S1.2 to sXcLive2450 produces an error.
7. P1 line 81 ‘joint effort’; line 83 ‘live cell’. Line 48: consider adding ‘but not mice’.
8. P2 line 98 ‘under a controlled’. Consider mentioning that the two labs were in Basel & Hannover. Line 132 ambiguous: ‘Tail intensity/% DNA in tail’ could read as a quotient – apparently ‘% DNA in tail’ is the only measure used from the CA data.
9. Fig 2 and others: the ‘exposed’ legend on the graph is filled but not coloured (to cater for the multiple colours used). Some mention of this in the Fig. caption might be useful.
10. Fig 3 KBrO3 squares in legend are indistinguishable: need to use another fill.
11. P 10 line 330 ‘temporary’?
12. P 11 line 371: hard to see the grey triangles, so this point is lost
13. P 14 line 507: ‘variables’.
14. P15 line 510: the evidence for making this statement is weak and the sentence needs to be modified: ‘to be caused possibly by a…’
Reviewer 2 Report
The Authors present very important issue – which is the influence of electromagnetic field with different parameters on: human MRC-5 lung fibroblast cells, human trophoblast cells HTR-8/SVneo and human osteosarcoma U-2 OS cells. Alkaline Comet Assay and Sister Chromatid Exchange assay were performed to determine genotoxicity. The authors have provided detailed data and analysis about performed experiments. Strictly defined conditions of in vitro studies made it possible to compare and give meaningful results.
- English correction is required, basically with respect to minor language errors, for instance:
“We conclude on the basis of our data that the possible carcinogenicity of wEMF modulations cannot be accounted for by their potential to…”
“Classified by IARC as “possibly carcinogenic to humans” (Group 2B), underlying molecular mechanisms have remained uncertain.”
„DNA insults” – I am not sure if this is scientific language.
- Abstract: “Classified by IARC as “possibly carcinogenic to humans” (Group 2B), underlying molecular mechanisms have remained uncertain.”
The Authors should explore more this statement about “uncertainly” in the Introduction. Are there none reports underlying mechanisms?
- Line 57: Comet Assay is a “ single cell gel electrophoresis”
- Why the Authors used these types of cells ? (besides of the references 22, 23 on which the Authors rely). Was there any particular reason? Primary human fibroblast MRC-5 and immortalized (non tumorigenic) HTR-8/SVneo ? While so much effort has been done, and well established methodology, particularly with regard to study tumorigenesis, maybe reasonable would be usage different cells, for example stem cells or cancer stem cells?
- The Authors are making an effort to validate their results with usage of H2O2, why The Authors have not performed longer exposure to see eventually the tendency in different cell types and with different parameters used. Why 1, 4, 24 hours?
- The Authors mentioned that: “In line with this, we have previously shown that primary human fibroblast cells only responded to a 50 Hz MF…”. Indeed there is number of research showing the influence of extremely low frequency electromagnetic field (ELF-EMF) on proliferation, death, cell cycle, hormone production, throphic activity and also genotoxicity. Were the Authors anyhow evaluating/calculating the parameters of magnetic field for in vitro experiments in terms of/ to the level of monolayers used in the experimental strategy?
- Some cited literature is relatively old. For example the obtained results might be the compensation of other processes i.e.:
https://www.ncbi.nlm.nih.gov/pmc/articles/PMC6701339/
- The manuscript is written in a bit chaotic way, the experimental strategy (in Materials and Methods) is not clear. It is suggested make it clear.
